# Current Trends in Clinical Trials for Merkel Cell Carcinoma (MCC)

**DOI:** 10.3390/cancers17142340

**Published:** 2025-07-15

**Authors:** Tilak Patel, Rachel Butz, Brian Boulmay, Vilija Vaitaitis

**Affiliations:** 1 Division of Hematology and Oncology, Department of Medicine, Louisiana State University Health Sciences Center New Orleans, New Orleans, LA 70112, USA; tpate2@lsuhsc.edu (T.P.); bboulm@lsuhsc.edu (B.B.); 2School of Medicine, Louisiana State University Health Sciences Center New Orleans, New Orleans, LA 70112, USA; rbutz@lsuhsc.edu; 3Department of Otolaryngology-Head and Neck Surgery, Louisiana State University Health Sciences Center New Orleans, New Orleans, LA 70112, USA

**Keywords:** Merkel cell carcinoma, cutaneous malignancy, clinical trials

## Abstract

Merkel cell carcinoma (MCC) is a rare and aggressive neuroendocrine cutaneous malignancy. Approximately half of patients diagnosed with MCC have their primary tumor located in the head and neck region adn the incidence is increasing due to an aging population and increased UV sun exposure. Traditional treatments have included surgery, radiation, and/or chemotherapy. However, new clinical trial studies are now testing the efficacy of immunotherapy, direct tumor injections, alternative radiation dosing and combination treatments to evaluate improvement in patient outcomes. This paper reviews the latest trends in MCC clinical trials research, along with potential future directions, in managing this rare and complex malignancy.

## 1. Introduction

MCC is a rare but aggressive neuroendocrine cutaneous malignancy, with approximately 1600 cases per year in the United States [1]. While MCC most frequently arises in the dermis, it can originate from any layer of the skin, appearing as a painless, nonulcerative, red-to-purple dermal papule [2]. Just under half (48.1%) of all MCC cases have a primary tumor located in the head and neck region, an area especially prone to sun exposure, a major risk factor for MCC [1]. Other risk factors include an age > 65 years, immunosuppression, and infection with Merkel cell polyomavirus (MCPyV) [1].

MCC is particularly concerning due to its 25–50% disease-related mortality, exceeding that of melanoma [2,3]. Factors independently associated with greater MCC mortality include the male sex, primary tumor location, tumor size, tumor depth, MCPyV-negative status, distant metastases, and nodal metastases [1]. MCC has a higher rate of metastatic spread than most other cutaneous malignancies [1]. Approximately 26–36% of MCC patients have clinical lymph node involvement at presentation, whereas fewer than 5% present with distant metastases [3]. Lymphovascular invasion (LVI) likely occurs early in MCC pathogenesis and is associated with a high incidence of microscopic nodal disease and worse prognosis [1].

The current first-line treatment recommendation for clinically node-negative (N0) MCC includes wide local excision (WLE) and sentinel lymph node biopsy (SLNB), followed by prompt postoperative adjuvant radiotherapy (RT) to the primary tumor bed and the lymph node basin if the SLNB is positive, surgical margins are close or positive, or when adverse features are present [1,2,4]. In addition to receiving adjuvant RT to the nodal basin, some patients undergo complete lymph node dissection (CLND); however, the use of RT for micrometastatic disease has been shown to have equivocal survival outcomes [1,5]. Since MCC has proven to be particularly radiosensitive, adjuvant RT has been shown to increase survival compared to CLND alone [1,5]. For stage IIIB MCC (clinically node-positive disease), multimodality therapy with WLE, CLND, and adjuvant RT to the primary and nodal sites increases 5-year overall survival to 30–40% [1]. Cytotoxic chemotherapies have also been used in non-metastatic MCC, but they are no longer recommended since they failed to improve overall survival [1]. However, treatment guidelines for MCC have recently expanded to include neoadjuvant treatment with immune checkpoint inhibitors, such as avelumab and pembrolizumab [1,3].

While SLNB is a good prognostic indicator for overall survival in non-head and neck MCC, variable lymphatic drainage complicates its ability to predict prognosis and the recurrence of head and neck MCC (HN-MCC) [3]. Even with a negative SLNB, HN-MCC carries a high risk of regional recurrence and mortality [3]. While adjuvant radiotherapy may assist in locoregional control, locoregional recurrence still occurs in 25–50% of all cases, contributing to poor outcomes [3]. For this reason, many high-risk patients are encouraged to participate in clinical trials [1].

This review provides a comprehensive analysis of current clinical trial trends in MCC management. We aim to explore the prognostic values of the various treatments for MCC, including surgical approaches, medical therapy (including chemotherapy and immunotherapy), and radiotherapy methods, based on ongoing clinical trials.

## 2. Recent Data Sets and Current Trials

Chemotherapy was historically the mainstay for advanced MCC, but recent retrospective analyses highlight its limited utility in the metastatic setting. Cytotoxic regimens (commonly a platinum-based regimen plus etoposide) can initially induce tumor regression, with first-line chemotherapy response rates being around 30–55% in MCC [6]. However, responses are typically short-lived: the median PFS after first-line chemotherapy is approximately 3–5 months, and virtually no patients achieve durable remission beyond one year [6]. Given this lack of durable benefit, chemotherapy’s role in MCC has diminished, and it is now reserved for patients who cannot receive immunotherapy. Recent trials in MCC have largely shifted away from chemotherapy in favor of immune-based therapies.

Immune checkpoint blockade is now considered the standard first-line therapy for unresectable and metastatic MCC. Programmed death-1 (PD-1) is an inhibitory receptor expressed on activated T cells, and its ligand PD-L1 is often upregulated on tumor cells. The binding of PD-1 to PD-L1 suppresses T-cell activity, allowing tumor immune evasion; PD-1/PD-L1 inhibitors restore T cell function and promote antitumor immunity by blocking this interaction. PD-1/PD-L1 inhibitors have demonstrated superior efficacy in both chemotherapy-refractory and frontline settings, with higher response rates and improved durability compared to chemotherapy [7,8].

Avelumab, an anti-PD-L1 antibody, became the first FDA-approved agent for metastatic MCC in 2017 after the phase II JAVELIN Merkel 200 trial (NCT02155647) showed tumor regressions in chemotherapy-refractory patients [7,9]. The trial confirmed an objective response rate (ORR) of 33% with a median response duration of 40 months, establishing avelumab as an effective long-term therapy option for MCC [7]. Similarly, pembrolizumab, an anti–PD-1 antibody, has demonstrated higher response rates than historical chemotherapy. In a multicenter phase II trial (NCT02267603), first-line pembrolizumab achieved an ORR of 56%, including 24% durable complete responses [8,10]. Another PD-1 antibody, nivolumab, showed efficacy in a phase I/II CheckMate 358 trial (NCT02488759), achieving a 60% ORR among patients with advanced MCC who were naïve to immune checkpoint inhibitors [11].

Recently, a randomized phase II trial evaluated first-line nivolumab plus ipilimumab with or without stereotactic body radiotherapy (SBRT) in advanced MCC. In ICI-naïve patients, the combination achieved a 100% ORR (41% complete responses). Even among patients previously treated with PD-1/PD-L1 inhibitors, the ORR was 31%. Notably, adding SBRT did not significantly improve response rates (72% with vs. 52% without SBRT). These findings suggest that dual PD-1 and CTLA-4 blockage is a highly active option in advanced MCC, both as initial therapy and after PD-1 failure [12]. Additionally, in the phase II POD1UM-201 trial (NCT03599713), the PD-1 inhibitor retifanlimab achieved a 52% ORR (18% complete responses) as a first-line treatment for metastatic MCC, leading to its approval by the FDA in 2023 [13].

MCC is associated with Merkel cell polyomavirus (MCPyV) in approximately 80% of cases. While MCPyV-positive tumors tend to have lower tumor mutational burden and distinct biology compared to virus-negative MCC, both subtypes respond to immune checkpoint inhibitors. Clinical trials have shown that PD-1/PD-L1 blockade is effective regardless of viral status, with some studies suggesting slightly higher response rates in virus-positive tumors, though differences are not statistically significant. Therefore, MCPyV status is not currently used to guide immunotherapy selection [14,15].

These immunotherapies have not only increased response frequency but also introduced durable remission, and even the possibility of cure, in advanced MCC. Encouragingly, responses to the PD-1/PDL-1 blockade are often maintained long-term. Across trials, most responders remain in remission at 1 year, with 2-year survival rates of around 60–70% having been noted in pooled analyses [13]. The safety profile of PD-1/PD-L1 inhibitors in MCC is generally consistent with other tumors, with immune-related adverse events being manageable in most cases. Across trials, <15% of patients needed to discontinue anti–PD-1/L1 therapy due to toxicity [13].

Overall, the PD-1/PD-L1 blockade now represents the cornerstone of systemic therapy for metastatic MCC, offering meaningful survival improvement and prolonged disease control in approximately half of patients. Ongoing clinical trials are exploring strategies to extend immunotherapy benefits and the primary or acquired resistance that affects the remaining 50% of patients [7].

Given the success of checkpoint inhibitors in advanced MCC, an active area of investigation is their role in the adjuvant setting with or without radiation for high-risk disease after curative surgery. MCC has a high recurrence rate even when treated aggressively at early stages, so effective adjuvant therapy could potentially improve outcomes. Initial trials using the CTLA-4 blockade in this setting were not encouraging. The randomized ADMEC trial (NCT02196961) compared adjuvant ipilimumab versus observation in completely resected MCC and found no significant improvement in disease-free survival (DFS) with ipilimumab. The trial was stopped early due to lack of benefit and significant toxicity [16]. This prompted a switch to evaluating the PD-1 blockade in subsequent studies.

The ADMEC-O trial (NCT02196961) is an open-label phase II trial evaluating adjuvant nivolumab versus observation in resected stage I–IV MCC. In this German multicenter study (*n*= 179), adjuvant nivolumab showed a trend toward improved DFS at 12 and 24 months, but interpretation is complicated by the high use of adjuvant radiation in the observation arm (74% of patients) compared to the nivolumab arm (50%) [16]. A final analysis of ADMEC-O was recently reported, showing a 10% absolute improvement in 2-year DFS, though a longer follow-up is needed to assess the impact on overall survival [16]. In the United States, a large phase III trial is ongoing to clarify the utility of adjuvant treatment. The STAMP trial (EA6174, NCT03712605) randomized patients with completely resected MCC (stages I–III) to pembrolizumab versus observation [17]. Co-primary endpoints are recurrence-free survival and overall survival, and standard adjuvant radiation was permitted in both arms. This trial has been completed, and its results will help determine whether adjuvant PD-1 therapy should become routine in treatment.

Additionally, a phase III study in Europe (ADAM, NCT03271372) is evaluating adjuvant avelumab in patients with nodal metastases after surgery and radiation [18]. While no checkpoint inhibitor has been approved for adjuvant MCC, these studies collectively represent an effort to extend the benefits of immunotherapy into earlier-stage disease.

Another major trend in MCC trials is the exploration of combination therapies, especially those integrating radiation or other modalities with systemic treatment. Given that roughly half of advanced MCC patients do not respond to PD-1/PD-L1 monotherapy [6], researchers are investigating whether combination approaches can enhance immunotherapeutic efficacy or overcome resistance. Radiation therapy is frequently used in MCC for local tumor control and has immunomodulatory effects that might synergize with checkpoint blockade. Historically, radiation therapy has been recommended for positive or close surgical margins, high risk features, positive sentinel lymph node biopsies, and advanced tumors with metastatic spread. Although evidence guiding optimal dosing is limited, the NCCN guidelines recommend 50–60 Gy in the adjuvant setting and 60–66 Gy in advanced, nonsurgical cases [19]. NCT05100095 is a phase II trial currently recruiting and investigating the response rate with hypofractionated radiation therapy in comparison to standard radiation therapy [20].

Several trials are examining immunotherapy plus radiation combinations. For example, a phase II study (NCT03304639) tested pembrolizumab with SBRT in advanced MCC [21]. In the adjuvant/localized setting, another trial (NCT04261855) is evaluating avelumab with external beam radiation and radioimmunotherapy using Lutetium-177—labeled somatostatin analog ([177 Lu]-DOTATATE) for patients with high-risk MCC [22]. In advanced nonresectable tumors, the CARTA Trial (A Phase II Single-Arm Clinical Trial Assessing Comprehensive Ablative Radiation Therapy With Avelumab in Unresectable and Metastatic Merkel Cell Carcinoma) is currently recruiting to use comprehensive ablative radiation therapy (CART) in combination with avelumab (NCT04792073) [23].

The role of intratumoral injections is another newer focus in cutaneous malignancy therapy. The NCCN guidelines recommend Talimogene laherparepvec (T-VEC) in the use of in-transit disease [4]. Intralesional T-VEC has shown promise in certain cases when patients have disease that is refractory to PD-(L)-1 or cannot receive immunotherapy [24]. The Dose Escalation Study of Neoadjuvant Intratumoral PH-762 for Cutaneous Squamous Cell Carcinoma, Melanoma, or Merkel Cell Carcinoma (NCT06014086) is evaluating the safety and tolerability of intratumoral injections of PH-762 prior to definitive surgical management. PH-763 is a RNAi molecule that inhibits the immune checkpoint PD-1 in tumors, slowing their growth. This trial performs four weekly intratumoral injections followed by surgical excision 2 weeks after the last injection (NCT06014086) [25].

## 3. Future Directions

Ongoing preclinical research is identifying novel molecular targets and pathways in MCC to enable next-generation therapies. For example, inosine monophosphate dehydrogenase 2 (IMPDH2), a rate-limiting enzyme in guanosine nucleotide biosynthesis, was recently found to be essential for MCC cell survival. IMPDH2 inhibition induces acute DNA replication stress and growth arrest in MCC. Combining an IMPDH2 inhibitor with an ATR kinase inhibitor caused synergistic replication stress in vitro and suppressed MCC tumor growth in vivo, suggesting a therapeutic strategy to exploit replication stress in MCC [26].

Another promising target is nicotinamide N-methyltransferase (NNMT), a metabolic enzyme overexpressed in many cancers, including MCC. Functional studies showed that NNMT promotes MCC cell proliferation and chemoresistance, while short hairpin RNA (shRNA)-mediated NNMT knockdown significantly reduced MCC growth and increased chemosensitivity [27]. Recent preclinical efforts highlight NNMT as a compelling target, with multiple classes of investigational NNMT inhibitors under development. These include macrocyclic peptides as allosteric inhibitors, alkene-linked bisubstrate inhibitors that mimic NNMT’s natural substrates, and esterase-sensitive prodrugs of bisubstrate inhibitors for improved cellular uptake and tumor-selective activation. Such diverse inhibitor classes underscore NNMT’s potential clinical value, providing multiple avenues for targeted therapy in MCC [28,29,30]

In parallel, upcoming trials are exploring innovative immunotherapeutic approaches beyond PD-1 monotherapy. One example is intratumoral innate immune activators such as IFx-Hu2.0, a novel agent designed to overcome primary resistance to checkpoint blockade. Early-phase studies of IFx-Hu2.0 demonstrated systemic antitumor immune responses in MCC, and a phase III trial combining intratumoral IFx-Hu2.0 with pembrolizumab in first-line metastatic MCC is planned to begin soon [31]. Additionally, multi-checkpoint inhibitor combinations are being tested to target alternative T-cell exhaustion pathways. For instance, a phase II trial (TRICK-MCC) is evaluating triplet therapy with PD-1, LAG-3, and TIM-3 antibodies in patients with advanced MCC that progressed after PD-1 inhibitor treatment. These approaches aim to improve outcomes for the subset of MCC patients who have not achieved durable responses with current immunotherapies [32].

## 4. Conclusions

Although a rare tumor, the incidence of MCC is increasing due to an aging population and increased UV sun exposure. Historically, the treatment of metastatic and advanced disease has been limited, but an improved understanding of the disease has allowed for novel treatment options and innovative clinical trial opportunities with the utilization of immunotherapies, alternative radiation dosing, and intratumoral injections. Overall, current trials reflect a broad effort to integrate systemic therapies with other treatments to improve patient outcomes.

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
