# Peer review of "Current Trends in Clinical Trials for Merkel Cell Carcinoma (MCC)"

_cancers, 2025, doi:10.3390/cancers17142340_

Round 1
Reviewer 1 Report
Comments and Suggestions for Authors
In the Opinion manuscript Current Trends in Clinical Trials for MCC. Patel et al address the historical, as well as current and ongoing treatment regimens for MCC.
Comments:
- Please include version of NCCN guidelines that are referenced, as treatment regimens are evolving.
- 2The authors cite a less recent article (Lango et al) referencing CLND as a surgical option for SLN with micrometastatic disease, which is not incorrect, but is actually being used less commonly - could also cite the paper from Penn by Ma et al (Ann Surg Oncol (2023) 30:4345-4355) that evaluated CLND and/or RT for this clinical presentation.
- The large clinical trail of combined Nivo and Ipi with/without stereotactic RT in advanced MCC should also be cited (Lancet. 2022 Sept 24; 400(10357): 1008-1019. This study addresses combined ICIs/RT as first line treatment or after failure of prior PD1/PDL1 in advanced MCC
- For intratumoral injections, should include that T-VEC is an option for N+ intransit metastasis per the NCCN v2.2025 guidelines.
- Please update any of the clinical trails cited to assure that preliminary results via ASCO abstracts or final publications are not available at the time of revision.
Author Response
Comment 1: Please include version of NCCN guidelines that are referenced, as treatment regimens are evolving.
- Thank you for this feedback. We have included the newest NCCN guidelines v2.2025 and have added it as a reference.
Comment 2: The authors cite a less recent article (Lango et al) referencing CLND as a surgical option for SLN with micrometastatic disease, which is not incorrect, but is actually being used less commonly - could also cite the paper from Penn by Ma et al (Ann Surg Oncol (2023) 30:4345-4355) that evaluated CLND and/or RT for this clinical presentation.
- Thank you for this comment. We agree with your feedback and therefore we have updated this in the paper and included the Ma et al paper in page 1, paragraph 4.
Comment 3: The large clinical trail of combined Nivo and Ipi with/without stereotactic RT in advanced MCC should also be cited (Lancet. 2022 Sept 24; 400(10357): 1008-1019. This study addresses combined ICIs/RT as first line treatment or after failure of prior PD1/PDL1 in advanced MCC
-
Thank you for this comment. We agree and therefore we have added this citation and information on page 3, paragraph 1.
Comment 4: For intratumoral injections, should include that T-VEC is an option for N+ intransit metastasis per the NCCN v2.2025 guidelines.
- Thank you for this comment. We agree and therefore we have added this citation and information on page 4, paragraph 4 along with citation for the paper the NCCN guidelines refer to.
Comment 5: Please update any of the clinical trails cited to assure that preliminary results via ASCO abstracts or final publications are not available at the time of revision.
- Thank you for this comment, it has been done.
Reviewer 2 Report
Comments and Suggestions for Authors
The manuscript “Current Trends in Clinical Trials for Merkel Cell Carcinoma (MCC)” is a short note about the latest trends in MCC clinical trials.
The manuscript is concise and well-written; few typos are present (e.g. double dots at the end of the simple summary). However, I suggest to address the following concern:
The manuscript lacks of a short paragraph about recent findings and possible future trials which would help to differentiate the present manuscript to already published similar manuscripts (PMID: 40412895). Although not already in clinical trials, several biomarkers have been investigated as potential therapeutic targets in MCC. These targets include for instance isozyme of de novo guanosine biosynthesis (PMID: 40491496) or nicotinamide N-methyltransferase (PMID: 38504052); for the latter there are available inhibitors which have been proposed for translational applications (macrocyclic peptides as allosteric inhibitors, alkene-linked bisubstrate inhibitors, and esterase-sensitive prodrugs of bisubstrate inhibitors).
Author Response
Comment 1: The manuscript lacks of a short paragraph about recent findings and possible future trials which would help to differentiate the present manuscript to already published similar manuscripts (PMID: 40412895). Although not already in clinical trials, several biomarkers have been investigated as potential therapeutic targets in MCC. These targets include for instance isozyme of de novo guanosine biosynthesis (PMID: 40491496) or nicotinamide N-methyltransferase (PMID: 38504052); for the latter there are available inhibitors which have been proposed for translational applications (macrocyclic peptides as allosteric inhibitors, alkene-linked bisubstrate inhibitors, and esterase-sensitive prodrugs of bisubstrate inhibitors).
- Thank you for this feedback. We agree and therefore have added several paragraphs in a Future Directions section at the end of the manuscript with comments on these potential future therapies.
Round 2
Reviewer 1 Report
Comments and Suggestions for Authors
Revisions sufficient for publication
Reviewer 2 Report
Comments and Suggestions for Authors
The authors properly addressed all the raised concerns and therefore the manuscript can be accepted for publication.